# Retrocopying expands the functional repertoire of APOBEC3 antiviral proteins in primates

Lei Yang[1], Michael Emerman[2], Harmit S Malik[3,4]*, Richard N McLaughlin Jnr[1,3]*

[1]Pacific Northwest Research Institute, Seattle, United States; [2]Division of Human Biology, Fred Hutchinson Cancer Research Center, Seattle, United States; [3]Division of Basic Sciences, Fred Hutchinson Cancer Research Center, Seattle, United States; [4]Howard Hughes Medical Institute, Fred Hutchinson Cancer Research Center, Seattle, United States

**Abstract** Host-virus arms races are inherently asymmetric; viruses evolve much more rapidly than host genomes. Thus, there is high interest in discovering mechanisms by which host genomes keep pace with rapidly evolving viruses. One family of restriction factors, the *APOBEC3 (A3)* cytidine deaminases, has undergone positive selection and expansion via segmental gene duplication and recombination. Here, we show that new copies of *A3* genes have also been created in primates by reverse transcriptase-encoding elements like LINE-1 or endogenous retroviruses via a process termed retrocopying. First, we discovered that all simian primate genomes retain the remnants of an ancient *A3* retrocopy: *A3I*. Furthermore, we found that some New World monkeys encode up to ten additional *APOBEC3G (A3G)* retrocopies. Some of these *A3G* retrocopies are transcribed in a variety of tissues and able to restrict retroviruses. Our findings suggest that host genomes co-opt retroelement activity in the germline to create new host restriction factors as another means to keep pace with the rapid evolution of viruses. (163)

*For correspondence:
hsmalik@fhcrc.org (HSM);
rmclaughlin@pnri.org (RNM)

**Competing interests:** The authors declare that no competing interests exist.

## Introduction

Host genomes have an ancient history of coevolution with selfish genetic elements. One type of these selfish elements, called endogenous retroelements, created a substantial fraction of most animal genomes (*Canapa et al., 2016*; *de Koning et al., 2011*; *Lander et al., 2001*; *Smit et al., 2015*; *Sotero-Caio et al., 2017*). Endogenous retroelements such as endogenous retroviruses (ERVs) and Long Interspersed Element-1s (LINE-1s) reside in host genomes where they 'copy-and-paste' themselves via the action of their reverse transcriptase. These retroelements can negatively impact host fitness by disrupting genes or regulatory regions, and by increasing the likelihood of ectopic recombination (*Boissinot et al., 2001*; *Hancks and Kazazian, 2012*; *Kaer and Speek, 2013*; *Petrov et al., 2003*; *Song and Boissinot, 2007*).

In addition to acting on their own RNA to ensure duplication, the reverse transcription/integration functions encoded by LINE-1s and ERVs also occasionally act on host mRNAs. This 'off-target' activity, termed retrocopying, entails the duplication of a host gene via the reverse transcription and integration of an mRNA. These 'retrocopies' are intronless and removed from the chromosomal location of the parental intron-containing gene. Previous studies estimated that 3,700–18,000 retrocopies are present in the human genome (*Casola and Betrán, 2017*; *Navarro and Galante, 2015*; *Potrzebowski et al., 2008*).

Two features distinguish retrocopies from other types of gene duplications. First, 'DNA-based' mechanisms of duplication (*e.g.*, segmental gene duplications) result in a new copy of the gene including its promoter and distal regulatory elements. In contrast, retrocopying typically duplicates

only the exons, leading to the moniker 'processed pseudogene'. Thus, transcription of a new retro-copy depends on the genomic neighborhood into which it integrates (*Carelli et al., 2016*). Second, retrocopying relies on the machinery encoded by endogenous retroelements like LINE-1, which are highly active in germline and early embryo (*Friedli et al., 2014*; *Garcia-Perez et al., 2007*; *Klawitter et al., 2016*; *Muotri, 2016*; *Wissing et al., 2012*). Therefore, unlike DNA-based duplications, retrocopying is almost exclusively limited to RNAs expressed in germline or early embryonic tissues. It follows that the level of germline expression of host mRNAs should be highly correlated to their probability of generating retrocopies. For example, ribosomal proteins that are highly expressed in germline tissues represent the most abundant class of processed pseudogenes in the human genome (*Balasubramanian et al., 2009*). While germline expression of host mRNAs predicates the generation of retrocopies, the vast majority of these retrocopies show characteristic signatures of pseudogenization (*Casola and Betrán, 2017*; *Navarro and Galante, 2015*; *Potrzebowski et al., 2008*).

While most retrocopies do not increase the genic capacity of the host due to inactivating mutations, a subset of retrocopies escaped mutational abrasion, presumably because they provide a selective advantage to the host. Indeed, evidence of functional retention in retrocopied sequences has been found in diverse organisms and includes functions such as novel subcellular localization of proteins (*Rosso et al., 2008*), neurotransmitter metabolism (*Burki and Kaessmann, 2004*), courtship (*Wang et al., 2002*), fertility (*Kalamegham et al., 2007*), and pathogen restriction (*Malfavon-Borja et al., 2013*; *Sayah et al., 2004*). Such functional retention may be particularly beneficial in the case of host defense genes, whose functional diversification is necessary for host genomes to keep pace with pathogens. For example, retrocopying of the *CypA* gene between coding exons of the *TRIM5* gene has created novel *TRIMCyp* fusion genes that can potently restrict retroviruses including HIV-1 (*Malfavon-Borja et al., 2013*; *Newman et al., 2008*; *Nisole et al., 2004*; *Sayah et al., 2004*; *Virgen et al., 2008*; *Wilson et al., 2008*). In a remarkable case of convergent evolution, retrocopying has created *TRIMCyp* fusion genes multiple times during primate evolution, further expanding and diversifying the *TRIM* gene family for retroviral defense (*Brennan et al., 2008*; *Virgen et al., 2008*). In other examples, mobile element and viral genes themselves have been retrocopied and domesticated for various functions including antiviral defense (*Best et al., 1996*; *Fujino et al., 2014*; *Malik and Henikoff, 2005*; *McLaughlin et al., 2014*; *Ito et al., 2013*; *Yan et al., 2009*).

Here, we investigated whether retrocopying may have similarly diversified another family of host defense genes: the *APOBEC3* (*A3*) cytidine deaminases. Although the common ancestor of placental mammals likely encoded three *A3* genes (*Münk et al., 2012*), this locus has recurrently expanded and contracted throughout mammalian evolution (*Ito et al., 2020*), including a dramatic expansion to seven paralogous genes in catarrhine primates (Old World monkeys and hominoids) followed by recurrent positive selection of this expanded gene set (*Bulliard et al., 2009*; *Compton et al., 2012*; *Duggal et al., 2011*; *Henry et al., 2012*; *McLaughlin et al., 2016*; *OhAinle et al., 2006*). We found that ancient and recent retrocopying has further diversified the already expansive *A3* gene repertoire in primates, adding as many as ten new *A3*s outside the well-studied' *A3* locus'. Our work uncovered an ancient *A3* born via retrocopying in the common ancestor of simian primates and a dramatic, ongoing history of *A3* retrocopying in New World monkeys (NWMs). Many of these NWM-specific *A3* retrocopies are expressed and some retain the capability to restrict retroviruses. Thus, retrocopied *A3*s have continually expanded host defense repertoires in primate genomes.

## Results

### *A3I*: an ancient *A3* retrocopy in simian primates

The *A3* locus of human, other hominoids, and Old World monkeys comprises seven clustered genes, *A3A-A3H* (*Jarmuz et al., 2002*; *OhAinle et al., 2006*; *Silvas and Schiffer, 2019*; *Figure 1A*). We undertook an analysis to search for any variation in this gene structure in other primate genomes. We used BLASTn with each of the seven human *A3* nucleotide sequence to query all sequenced primate genomes on NCBI. As expected, each genome contained a series of proximal hits, presumably comprising the *A3* locus. However, in every simian primate genome examined, we also found exactly one shared syntenic region, distinct from the *A3* locus, with high similarity to *A3G*. This sequence

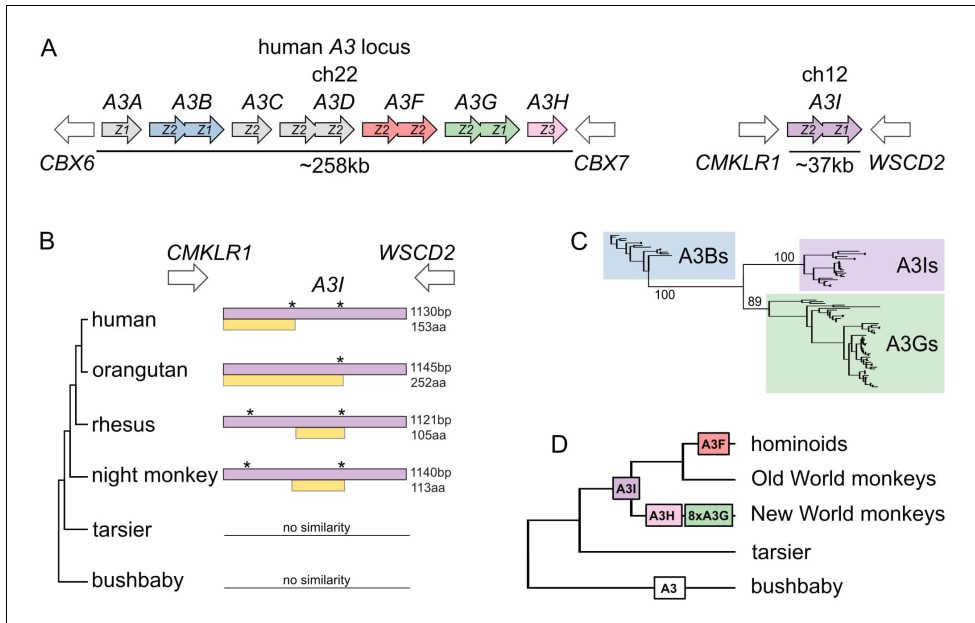

**Figure 1.** Identification and phylogenetic distribution of *A3I*. (**A**) *A3I* is located away from the *A3* locus at a distant but highly conserved syntenic locus in all simian primates. The human genome is shown as an example. (**B**) ORF structure of *A3I* in various primate species. Purple boxes represent sequences that can be aligned to the intron-containing *A3* copies, whereas yellow boxes represent the longest ORF of the *A3I* in corresponding species. Stars (*) indicate the position of stop codons. (**C**) Maximum likelihood phylogeny of *A3Is* and the intron-containing *A3Bs* and *A3Gs*. Clusters of *A3Is*, *A3Gs*, and *A3Bs* are highlighted by their respective color, and bootstrap values leading to these clusters are shown on the nodes. (**D**) Expansion of *A3* retrocopies along the primate phylogeny. The number of retrocopies of each A3 is shown in color boxes at the inferred point of retrocopy birth in the primate phylogeny. The white 'A3' box represents a sequence that could not be assigned to a particular ortholog. The online version of this article includes the following figure supplement(s) for figure 1:

**Figure supplement 1.** A phylogeny of the domains of primate *A3s* and *A3G* retrocopies A PhyML tree of the domains of *A3s* and *A3Is*.

**Figure supplement 2.** A phylogeny of primate *A3s* and *A3G* retrocopies A PhyML tree of nucleotide sequences of an alignable region of *A3Gs* and *A3Bs* from diverse simian primates in addition to *A3G* retrocopies from New World monkeys.

match spanned the exonic sequences of *A3G* in a single contiguous region of around 1,100 bp (*Figure 1B*). Based on the absence of introns, we concluded that this sequence represents a retrocopy of an *A3* gene. We found no evidence of the syntenic copy of this *A3* retrocopy in the genomes of prosimians including the tarsier, bushbaby, and mouse lemur. We, therefore, conclude that this retrocopy was born in the common ancestor of simian primates, and hence propose the name *A3I* for this retrocopy, extending the nomenclature scheme for other human *A3* genes.

To understand the relationship between *A3I* and other *A3* genes, we created a maximum likelihood tree using the nucleotide sequences from simian primate *A3B*, *A3G*, and *A3I* genes. *A3I* could be aligned to primate *A3Gs* and *A3Bs*, since *A3I* shares the deaminase domain organization of these *A3s* (*A3Z2-A3Z1*; *Figure 1A*). With high bootstrap support, we found that all *A3Is* share a common ancestor to the exclusion of *A3B* and *A3G* genes. This pattern held with a tree of individual deaminase domains from all human *A3s* (*Figure 1—figure supplement 1*). Our analyses found the *A3G* genes to be the closest phylogenetic neighbors of the *A3Is*, suggesting a common ancestry of these two genes (*Figure 1C*). Since *A3G* is predicted to have been born soon after the simian-prosimian split (*Münk et al., 2012*), we propose that *A3I* arose via retrocopying of *A3G* in the common ancestor of simian primates, approximately 43 million years ago (MYA) (*Perelman et al., 2011*).

In all species analyzed, *A3I* has acquired potentially inactivating mutations relative to *A3G*. We found that all *A3I* retrocopies share a nonsense mutation at codon position 261 (*Supplementary file 3*), and thus, in most species, *A3I* encodes only a short putative open reading frame (ORF) of 153

codons (compared to the 384 codon ORF of *A3G*) which spans the N-terminal deaminase domain. Following this initial truncation, there were additional lineage-specific disruptions of the *A3I* ORF during the diversification of simian primates. These results suggest that *A3I* was born in the simian ancestor either as a truncated retrocopy or acquired a truncating mutation shortly following birth. Nonetheless, it is possible that the ancient *A3I*s encoded functional A3 proteins before becoming disabled by mutation.

## Multiple *A3G* retrocopies in new world monkeys

*A3I* was not the only hit uncovered by our search for *A3* retrocopies. Our analyses also revealed *A3F* retrocopies in three Old World monkeys within the Colobinae subfamily, *A3H* retrocopies in two New World monkeys, and a single *A3* retrocopy that cannot be assigned to a specific *A3* parent in the greater galago (*Otolemur garnettii*) (*Figure 1D*). However, our most striking finding was that every sequenced NWM genome contained numerous *A3G* retrocopies. This abundance of *A3* retrocopies motivated a deeper investigation of the evolution and function of these NWM *A3G* retrogenes.

Initially focusing on the common marmoset (*Callithrix jacchus*) genome, we found three intron-containing *A3* genes on chromosome 1 (likely orthologous to human *A3A*, *A3G*, and *A3H*). We also found nine loci outside of the *A3* locus with high sequence similarity to human and marmoset *A3G* genes (*Figure 2A*). In contrast to the marmoset *A3G* gene, each of these additional hits lacked introns suggesting they were retrocopies. Seven *A3G* retrocopies spanned more than 1,100 bp and most of the coding exons of the marmoset *A3G*. In addition, one retrocopy spanned 811 bp, and one shorter retrocopy spanned 260 bp (*Figure 2A*). These shorter retrocopies showed a marked 3' bias, consistent with the 5'-truncation-prone, target-primed reverse transcription mechanism of

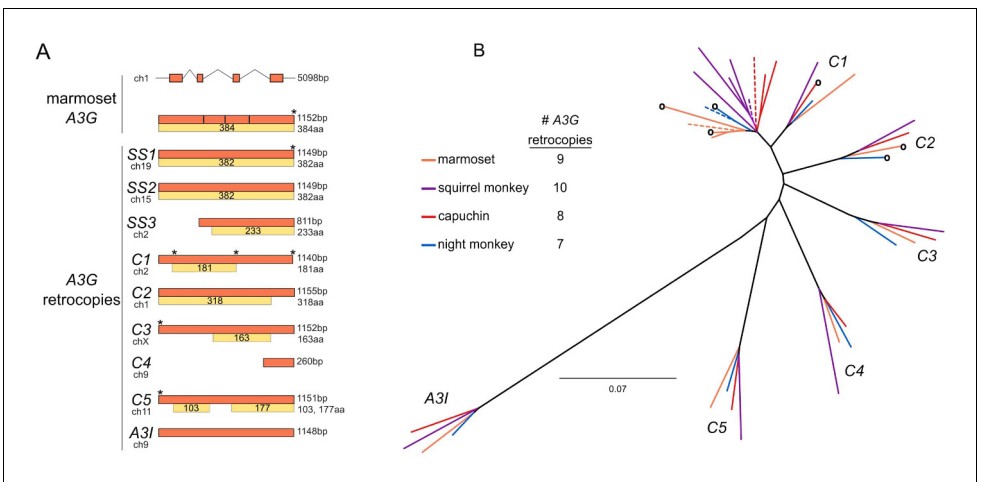

**Figure 2.** Discovery and phylogenetic analysis of *A3G* retrocopies in New World monkeys. (**A**) The common marmoset (*Callithrix jacchus*) genome encodes a single *A3G* and nine retrocopies of *A3G* (orange boxes). *A3G* resides on four coding exons at the *A3* locus on chromosome 1, while the retrocopies are intronless and found throughout the genome. Some retrocopies contain putative protein-coding ORFs (yellow boxes) of varying lengths that retain alignable sequence similarity to the *A3G* protein (gray within yellow boxes, regions of poor alignment caused by frame-shifting mutations). (**B**) PhyML tree of *A3G* and *A3G* retrocopies from the four sequenced and assembled New World monkey genomes suggests that six retrocopies are orthologous and conserved in all four species (clusters *C1-C5* and *A3I*). The genome of each species (colors correspond to species) contains an intron-containing *A3G* (dotted line) as well as retrocopies that are closely related to *A3G*. These more recent copies are found in only one genome, without identifiable orthologs in the other three species. Some retrocopies retain a putative protein coding ORF (indicated by a circle at the branch tip).

The online version of this article includes the following figure supplement(s) for figure 2:

**Figure supplement 1.** Synteny analysis of retrocopies in marmoset and squirrel monkey for inference of orthology UCSC table browser was used to identify the genes on either side of each retrocopy in marmoset and squirrel monkey.

**Figure supplement 2.** PCR of genomic DNA of New World monkeys to date retrocopy births.

LINE-1 and related retrotransposons (*Cost et al., 2002*; *Luan et al., 1993*). Seven *A3G* retrocopies possessed premature stop codons, deletions, or early truncations (*Figure 2A*). Two of these retrocopies encode ORFs that span a single cytidine deaminase domain ('*C1*' and '*C2*'). However, two *A3G* retrocopies ('*SS1*' and '*SS2*', *Figure 2A*) were predicted to encode a 382 amino acid protein (comparable to the full-length 384 amino acid protein encoded by the intron-containing *A3G* gene). Thus, the marmoset genome contains nine *A3G* retrocopies, which may encode 2–4 additional A3G-like proteins.

Next, we expanded our search for *A3G* retrocopies to the other assembled NWM genomes on NCBI: the Bolivian squirrel monkey (*Saimiri boliviensis*), the white-faced capuchin (*Cebus capucinus*), and Ma's night monkey (*Aotus nancymaae*). Like marmoset, we found multiple *A3G* retrocopies in each of these genomes – eight in capuchin, seven in night monkey, and ten in squirrel monkey (*Figure 2A*). All but one of the NWM *A3G* retrocopies aligned (without large gaps) to each other and intron-containing *A3G*s, and a phylogenetic tree confirmed these retrocopies cluster with NWM *A3G*s (*Figure 1—figure supplement 2*). The exception, one squirrel monkey *A3G* retrocopy (Gen-Bank: JH378161), contains a 286 bp insertion of another shorter *A3G* retrocopy, most likely the result of a nested insertion of one retrocopy into another.

A maximum likelihood phylogenetic tree (*Figure 2B*) using the alignable NWM *A3Gs* and *A3G* retrocopies revealed six bootstrap-supported 'clusters' of retrocopies with representatives from all four analyzed NWM species (*Figure 2B* and *Supplementary file 1* clusters C1-C5 and *A3I*). One of these clusters contains the NWM *A3I* sequences which date back to at least the last common ancestor of simian primates. Our findings suggest that the other five clusters represent orthologs of *A3G* retrocopies born in or before the last common ancestor of these four NWM species. This orthology was further supported by shared synteny in two NWM genomes (marmoset and squirrel monkey) for all clusters (*Figure 2—figure supplement 1*). We, therefore, conclude that five orthologous NWM *A3G* retrocopies were likely born via independent retrotransposition events in or prior to the most recent common ancestor of these four species analyzed.

To more precisely date the origins and species distribution of these *A3G* retrocopies, we investigated their presence in additional NWM species lacking publicly available genome sequences. For each retrocopy, we used UCSC MultiZ alignments (*Blanchette et al., 2004*) to find flanking sequence conservation in shared syntenic locations of marmoset, human, and mouse genomes to design oligos specific to a single retrogene-containing locus in marmoset. We were able to do so for seven of nine retrocopies (all but *SS3* and *C4*, *Figure 2A*). Confirming the specificity of these oligos, six of these seven oligo pairs reproducibly amplified a single locus from marmoset genomic DNA with touchdown PCR whereas only *A3I* was amplified from human genomic DNA (*Figure 2—figure supplement 2*). Using these oligos and genomic DNA from other species, we observed that many retrocopies were present in the shared syntenic loci in other NWMs. Specifically, retrocopies *C1*, *C2*, *C3*, and *C5* of marmoset were also present in titi (*Plecturocebus moloch*) and saki monkeys (*Pithecia pithecia*), two species in the basal family of the NWM phylogeny, suggesting these retrocopies were born in or prior to the most recent common ancestor of all NWMs (~25 MYA).

In addition to the six orthologous 'clusters' of retrocopies found in all or many NWMs, our phylogenetic analysis also reveals 'species-specific' retrocopies (*SS*) with no apparent ortholog in the other three species with genome assemblies (*Figure 2B*). These *A3G* retrocopies instead share a recent common ancestor with the intron-containing *A3G* gene from the same species, suggesting that they were born recently. Our PCR analysis revealed that some 'marmoset-specific' retrocopies were also present in the closely related tamarin (*Saguinus oedipus*). Thus, *A3G* retrocopies vary in age, from being found in only one species, in a few closely related species, in all NWM species, or in all simian primates. The different branch lengths leading to each of the NWM *A3G* retrocopies or retrocopy ortholog clusters also reflect their variable ages (*Figure 2B*). Our findings suggest that rather than a single burst, *A3G* retrocopies have been continually born throughout the evolutionary history of NWMs.

## Retention of putatively functional NWM *A3G* retrogenes

Retrocopies are often assumed to be nonfunctional at birth since they usually consist of only the sequence within the mRNA of the parent gene and therefore lack promoters, and enhancers. However, there are well-documented examples of retrogenes that have been retained for their functionality (*Casola and Betrán, 2017*). To investigate whether any of the *A3* retrocopies might be

functional, we used several criteria to eliminate retrocopies likely to be non-functional. We assumed that functional retrocopies should be transcribed and should have evolved under selective constraint on an intact open reading frame with an intact cytidine deaminase motif. To be conservative, we narrowed our focus to the *A3G* retrocopies detectable in the four sequenced, publicly available New World monkey genomes – nine copies in common marmoset, eight copies in white-faced capuchin, ten copies in Bolivian squirrel monkey, and seven copies in Nancy Ma's night monkey.

Many *A3* genes are expressed in the germline and early development (*Friedli et al., 2014*; *Marchetto et al., 2013*; *Refsland et al., 2010*) where they protect against a diverse range of infectious and endogenous elements including retroviruses and LINE-1 (*Arias et al., 2012*; *Harris and Dudley, 2015*). In order to similarly assess expression of *A3G* retrocopies in vivo, we queried publicly available NWM RNA-seq datasets (*Supplementary file 2*) using all reference *A3G* retrocopy sequences. We organized all perfect-matching, uniquely-mapping read counts by species (*Figure 3*, x-axis) and specific retrocopy (*Figure 3*, y-axis). The intron-containing *A3G* gene itself showed detectable expression in most datasets in all four species (organized by tissue source of each dataset, colored bars down columns denote tissues). We also found that several *A3G* retrocopies showed expression in each species (dark bars within each species). For example, *C2* retrocopies (*Figure 3*) are expressed in three of the four analyzed NWM species. In marmoset, the *C2* retrocopy is expressed in stem cells and induced pluripotent cells, similar to the intron-containing *A3G* gene. We also found that each species expressed at least one species-specific retrocopy. In several species, these younger, species-specific *A3G* retrocopies were expressed in ovaries and testes just like the intron-containing *A3G*. Overall, our data suggest that a large subset of NWM *A3G* retrocopies are expressed in vivo, including in tissues relevant for defense against pathogens.

Second, we evaluated the *A3G* retrocopies for their predicted ORFs. Full-length NWM *A3G* genes encode a ~ 384 amino acid protein. In contrast, most *A3G* retrocopies encode only short putative ORFs, typically less than 100 codons (*Figure 2A*, yellow boxes). However, a subset of *A3G* retrocopies have retained a predicted ORF of at least 250 codons (*Figure 2A*, yellow boxes; *Figure 2B*, empty circles on branches) which encompasses one or two of the core deaminase domains. Most of these retrocopies conserve the core amino acids within these domains required

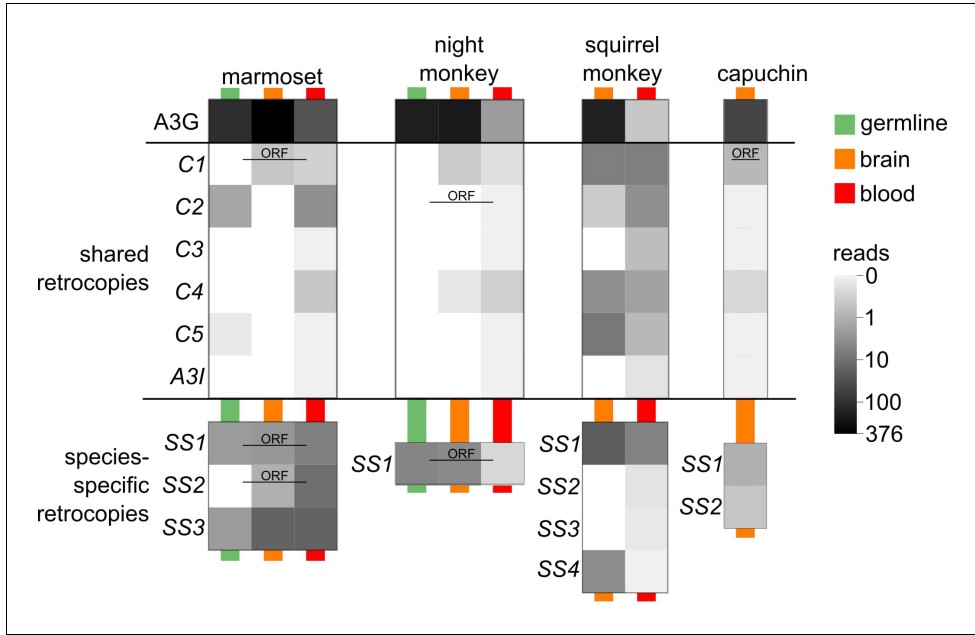

**Figure 3.** *A3G* retrocopies are transcribed in New World Monkey tissues A heat map shows the counts of RNA-seq reads ($\log_{10}$ of read count + 1) that map uniquely at 100% identity and coverage. Each pixel represents the average read counts of available data for the corresponding tissue type and *A3G* retrocopy. Tissue types are marked by the colored lines behind the pixels. Green represents germline tissues including iPSC, ESC, testis and ovary; orange represents brain tissues of various regions; red represents blood samples including whole blood and lymphocytes. Retrocopies which retain a putative protein coding ORF are labeled with 'ORF'.

for antiviral or anti-retroelement activities of *A3G* (*Figure 4*; *Huthoff and Malim, 2007*; *Navarro et al., 2005*).

Third, we evaluated the *A3G* retrocopies for evidence of selective retention. Although *A3G* retrocopies are expected to lack stop codons upon their birth from an intact *A3G* gene, absence of stop codons in older *A3G* retrocopies could indicate functional retention. We adapted a previously published approach (*Young et al., 2018*) to simulate the rate of decay of ORFs in the absence of selection based on ORF length and conservative and liberal bounds on NWM background mutation frequency and generation time (*Campbell and Eichler, 2013*; *Tacutu et al., 2018*; *Thomas et al., 2018*). We found that that less than 5% of *A3G* ORFs were expected to remain intact after 20 million years (less than 1% after 40 million years) (*Figure 5*). In contrast to this expectation, we found two *A3G* retrocopies have remained intact despite being at least 20 million years old (*Bininda-Emonds et al., 2007*). These include one *C1 A3G* retrocopy (with a preserved ORF in capuchin monkey) and a *C2* retrocopy (with a preserved ORF in marmoset and night monkey). Based on these findings, we hypothesize that some NWM *A3G* retrocopies have been retained for their function.

To further evaluate the selective constraint acting on *A3G* retrocopies, we used computational models to test whether their evolution more closely resembles a functional gene or a pseudogene. We first used the RELAX method (*Wertheim et al., 2015*) to test whether the *A3G* retrocopies show relaxed selection relative to intron-containing *A3G* genes. Significant relaxed selection was not detected in the putatively intact retrocopies relative to the intron-containing *A3G*s (*Figure 5—figure supplement 1*). Instead, RELAX suggests that the *A3G* retrocopies have evolved more rapidly than the intron-containing *A3G* genes in the same set of species (*Figure 5—figure supplement 1*). Next, using a branch model of PAML (*Yang, 2007*), we observed that two retrocopies (capuchin-*C1* and marmoset-*SS1*) had elevated dN/dS (2.6 and 2.9 respectively, significantly greater than the neutral expectation of 1), while the rest of the branches were suggestive of neutral evolution or purifying selection (*Figure 5—figure supplement 1*). These analyses suggested that, overall, the retrocopies evolved at a similar or accelerated rate compared to intron-containing *A3G*s. Further, capuchin-*C1* and marmoset-*SS1* show evidence of accelerated evolution. Overall, our three lines of evidence suggest that at least a subset of the *A3G* retrocopies are likely to have been retained for their function.

## Antiviral activity of NWM *A3G* retrocopies

We reasoned that *A3G* retrocopies could have a role in innate immunity/genome defense similar to intron-containing *A3* genes. To test this possibility, we cloned and assayed intron-containing *A3G*s and each *A3G* retrocopy encoding an intact near-full-length ORF for its ability to restrict the

|  | N-terminal deaminase motif | | C-terminal deaminase motif | |
|---|---|---|---|---|
|  | HXE | CXXC | HXE | CXXC |
| human A3G | HPE | CTKC | HAE | CFSC |
| rhesus A3G | HPE | CTRC | HAE | CFSC |
| marmoset A3G | HPE | CPVC | HAE | CFCC |
| squirrel monkey A3G | HPE | CPVC | HAE | CFSC |
| owl monkey A3G | HPE | CPVC | HAE | CFSC |
| marmoset-SS1 | HPE | CPVC | HAE | CFSC |
| marmoset-SS2 | NPE | CPVC | HAE | CFSC |
| marmoset-C1 | HPE | CPVC | HAE | CFSC |
| night monkey-C1 | HPE | CPVC | HAE | CFSC |
| night monkey-SS1 | HPE | CPVC | HAE | CFSC |
| capuchin-C1 | HPE | SPVC | HAE | CFSC |

(retrocopies: marmoset-SS1 through capuchin-C1)

**Figure 4.** *A3G* retrocopies retain core deaminase motifs. An amino acid alignment of the core deaminase motifs shows that *A3G*s of various primates have conserved HxE-CxxC motifs in both the N- and C-terminal domains. The putative ORF-encoding retrocopies all retain a conserved C-terminal motif, and most retain an N-terminal motif.

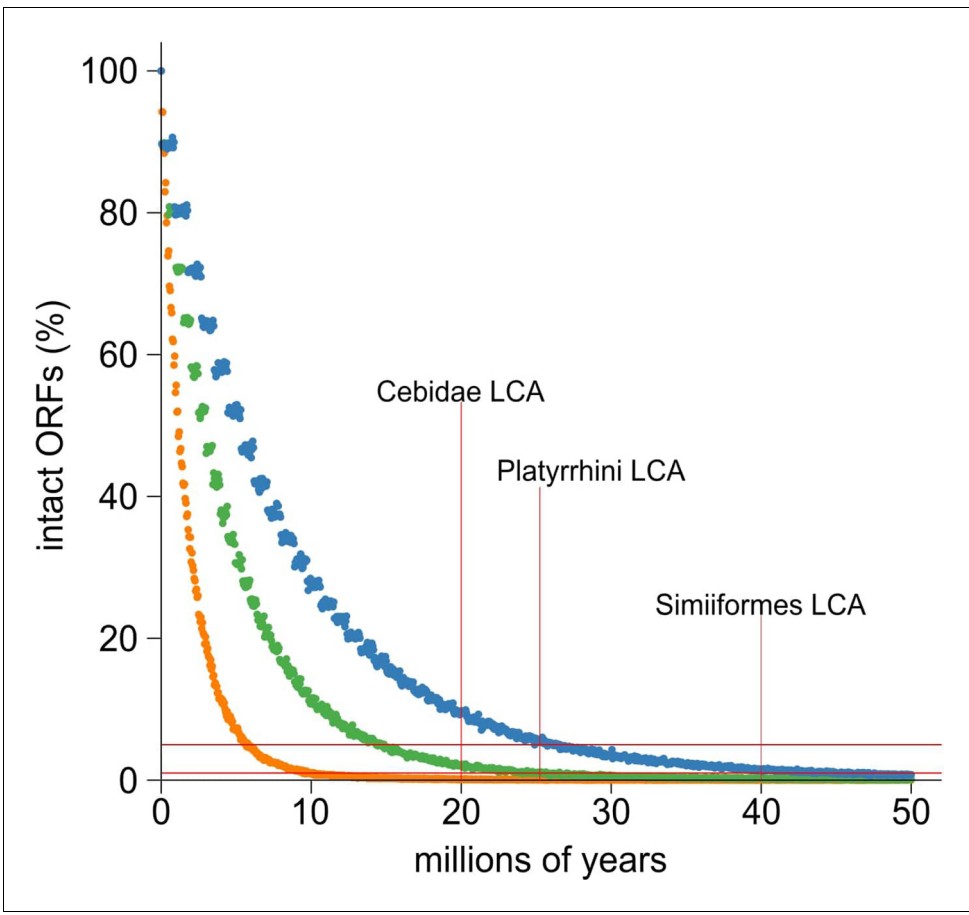

**Figure 5.** Simulation and evolution suggest selection to retain ORFs in *A3G* retrocopies. (**A**) A simulation of ORF retention suggests most are lost within 10–20 million years in the absence of any selection to retain the ORF. Dots indicate the proportion of simulated ORFs (10,000 total) that were still intact after a given time. Colors represent three sets of parameters intended to match New World monkeys (green) or provide liberal (orange, mouse-like) and conservative (blue, human-like) bounds on the parameter sets of indel rate and generation time. The substitution rate of Ma's night monkey was used for all three sets of simulations. Horizontal red lines indicate the 1 st and 5th percentile of intact ORFs. Vertical red lines mark the key time points of last common ancestors (LCA) among New World monkeys.

The online version of this article includes the following figure supplement(s) for figure 5:

**Figure supplement 1.** Analysis of selection in the evolution of retrocopies.

---

endogenous retroelement LINE-1 using established in vitro retrotransposition assays (*Dewannieux et al., 2003*; *Moran et al., 1996*). These assays require that a LINE-1 sequence be transcribed, spliced, and reverse transcribed back into the genome. As controls, we tested the anti-LINE-1 restriction of human A3A and human A3G. Consistent with previous reports (*Bogerd et al., 2006*; *Chen et al., 2006*; *Muckenfuss et al., 2006*; *Niewiadomska et al., 2007*), we observed potent restriction of LINE-1 by human A3A, and no restriction by human A3G. In contrast to human A3G, we found that the intron-containing A3Gs from marmoset and squirrel monkey restricted LINE-1 more than 10-fold, comparable to A3A. However, we observed no appreciable restriction of LINE-1 by any of the *A3G* retrocopies (*Figure 6*; *Figure 6—figure supplement 1*). Thus, despite potent anti-LINE-1 restriction by NWM A3Gs, it appears that this activity is not retained by any of the retro-copies tested.

Next, we investigated the antiviral restriction by NWM *A3G* genes and retrocopies. Using single-cycle infectivity assays, we measured the ability of NWM *A3G* genes and retrocopies to block infectivity of HIV-1ΔVif, which lacks Vif, a known antagonist of APOBEC3 proteins. Consistent with previous results, we found that human A3G potently restricts HIV-1ΔVif but human A3A is a poor

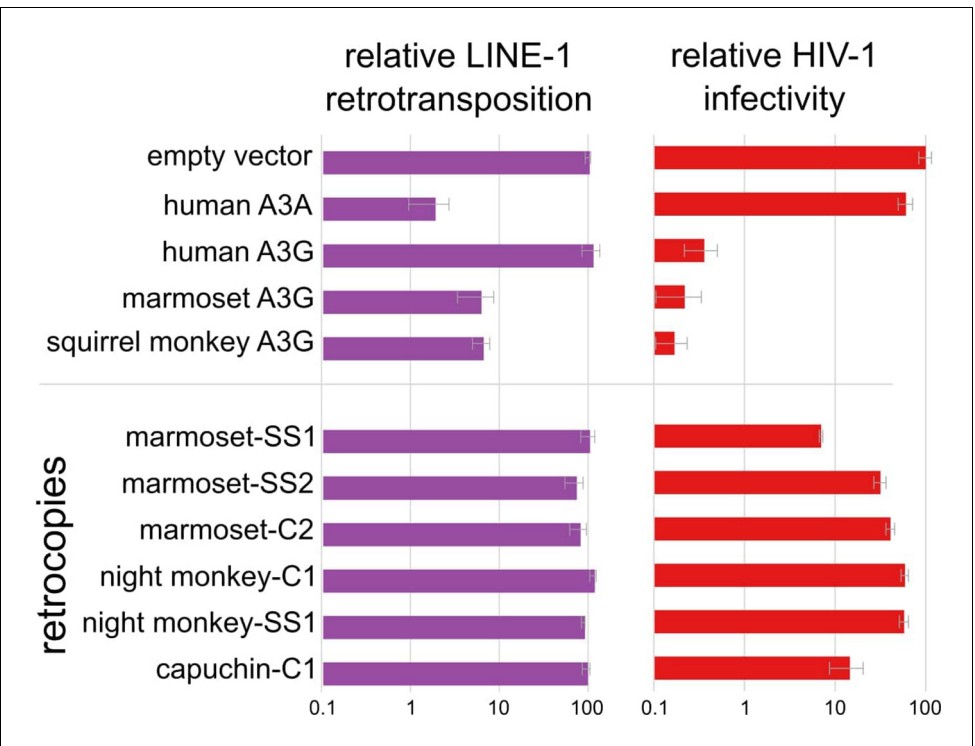

**Figure 6.** *A3G* retrocopies restrict HIV-1 but not LINE-1 Bar charts of measured restriction of LINE-1 (retrotransposition assays) and HIV-1ΔVif (single cycle infectivity assays) show that NWM *A3Gs* and some *A3G* retrocopies restrict retrovirus. Only NWM *A3Gs*, but not retrocopies restrict LINE-1.

The online version of this article includes the following figure supplement(s) for figure 6:

**Figure supplement 1.** Western blot of A3s and A3G retrocopies For each construct, 50 ng plasmid was transfected into 25,000 293T cells in a single well of a 24 well plate.

---

restrictor; this restriction pattern is the opposite to that observed for LINE-1 restriction in here and in previous findings (*Bogerd et al., 2006*; *Chen et al., 2006*; *Turelli et al., 2004*; *Figure 6*). We also observed 100-fold or greater restriction of HIV-1ΔVif infectivity by intron-containing *A3G* genes from marmoset and squirrel monkey (*Figure 6*; *Figure 6—figure supplement 1*), consistent with a previous report of restriction by NWM A3Gs (*Wong et al., 2009*). Finally, we observed that two retrocopies – marmoset-*SS1* and capuchin-*C1* – restrict HIV-1ΔVif at least 10-fold, suggesting that these two *A3G* retrocopies encode *bona fide* A3G-like anti-retroviral activity. Thus, retrocopying has expanded the functional repertoire of *A3* antiviral genes in NWMs. At least 2 of these genes are expressed at the RNA level in at least some tissues and encode a functional protein with antiviral activity.

## Discussion

Replicating retrotransposons inflict deleterious consequences on host genomes via insertional mutagenesis, ectopic recombination, and dysregulation of proximal genes (*Beck et al., 2011*). Despite these negative consequences, retrotransposons can bring about innovation in host genomes via the birth of new exons or genes (*Mi et al., 2000*; *Schmitz and Brosius, 2011*), or novel regulatory mechanisms and gene-regulatory networks (*Chuong et al., 2016*; *Kunarso et al., 2010*; *Wang et al., 2007*). In this work, we show that retrotransposon-mediated gene birth can lead to continual evolution of new innate immune genes. We show that all simian primate genomes contain the remnants of *A3I*, an ancient *A3* retrocopy. We further find that NWM genomes have continually acquired *A3G*-derived retrocopies, a subset of which are transcribed, retain intact ORFs and functional motifs, and are capable of restricting retroviruses.

This history of ancient and young retrocopies provides a valuable resource in understanding how antiviral genes coevolve with pathogens, including changes in Vif-interacting residues or viral restriction profile (*Krupp et al., 2013*). Although numerous methods exist for reconstruction of ancestral sequences, rapidly evolving genes like the *A3s* violate assumptions and often limit the utility of these methods, thereby preventing reliable reconstruction of ancestral sequences. However, retrocopies are molecular fossils, an evolutionary snapshot of the ancient parental gene sequence which presumably evolved neutrally after inserting into the genome. *A3I* provides such a record of an *A3G*-like gene from 40 MYA, which was present in the common ancestor of simian primates. Given the rapid gene turnover of the *A3* locus in mammals, it is possible that the parent of *A3I* no longer exists in modern primates. In this scenario, the *A3I* retrocopy may be all that remains of this ancient *A3* gene which predates simian primate diversification.

Recent computational analysis corroborates the presence of *A3* retrocopies in two of the genomes we analyzed (*Hayward et al., 2018*; *Ito et al., 2020*) and adds to a growing literature suggesting the *A3* content of mammalian genomes may be even more variable and dynamic than previously appreciated (*Hayward et al., 2018*; *Ito et al., 2020*). Our data suggests that *A3* retrocopying is more prevalent in NWM genomes compared even to other simian primates. This abundance is consistent with a previous study that reported an increased number of retrocopies of all genes in marmoset and squirrel monkey genomes, correlated with an increase in the activity of two LINE-1 subfamilies L1PA7 and L1PA3 (*Navarro and Galante, 2015*). It is unclear whether increased LINE-1 activity is sufficient to explain our observations since some NWMs like the *Ateles* lineage may have low or no retroelement activity (*Boissinot et al., 2004*). Even if NWM LINE-1 activity is high, it would not necessarily explain why *A3G* rather than the other NWM *A3* genes are subject to recurrent retrocopying. Although duplication of some nuclear A3 proteins like human A3A or A3B are likely to be more toxic due to increased genomic mutation (*Hultquist et al., 2011*; *McLaughlin et al., 2016*), we favor the alternate hypothesis that *A3G* expression in the germline/early embryos of NWMs is unusually high, rendering it a more likely substrate for retrocopying relative to other NWM *A3* genes. Following their insertion into a new genomic location, these retrocopies could be expressed by exaptation of a neighboring transposable element, promoter piggybacking, or recruitment of a novel promoter (*Carelli et al., 2016*). Recent work suggests that most of the mouse genome is transcribed over relatively short evolutionary timescales (*Neme and Tautz, 2016*). Such 'genome-wide' transcription could be the first step in exposing an advantageous function of a retrogene (*Jaganathan et al., 2019*).

We showed that intron-containing NWM *A3G* genes restricted both LINE-1 and HIV-1. Thus, it is likely that *A3G* retrocopies retained both of these functions immediately following birth. Yet, over time, all retrocopies that restrict HIV-1 have lost the ability to restrict LINE-1 (*Figure 6*). Although this could reflect idiosyncratic events, our finding that anti-LINE-1 activity, but not anti-retroviral activity, was repeatedly lost, suggests otherwise. It is possible that the anti-LINE-1 function is simply more sensitive to random mutation, such that mutations are more likely to result in loss of LINE-1 restriction; we also cannot rule out the possibility that certain *A3G* retrocopies retain the capacity to restrict NWM-specific LINE-1 lineages. Alternatively, *A3G* retrocopies may have been absolved of selection for LINE-1 restriction, perhaps due to sufficient silencing by *A3G* and other restriction factors. Nevertheless, the retrocopies present a natural 'separation of function' event that can delineate the requirements for *A3G* proteins to restrict LINE-1 versus retroviruses.

Although we used the lentivirus HIV-1ΔVif to measure the anti-retroviral activity of NWM *A3G* retrocopies, lentiviruses have not yet been found in NWMs. Even apart from lentiviruses, few active retroviruses in general have been found in NWMs; those that have been found likely represent the tip of an understudied aspect of monkey and virus biology (*Colcher et al., 1977*; *Muniz et al., 2013*). Thus, HIV-1ΔVif only serves as a proxy for the activity of *A3G* retrocopies towards some relevant viral pathogen in the natural environment of these monkeys.

While the NWM *A3G* retrocopies did not restrict LINE-1, such a mechanism of gene duplication could, in theory, function as a feedback mechanism on excess retroelement activity in the germline/early embryo. Retroelement restriction factors expressed in these tissues could be retrocopied and increase dosage or diversity of anti-retroelement restriction factors (*Kondrashov et al., 2002*). In this way, the retrocopies may represent a 'revolving door' of new gene substrates for neo- or subfunctionalization; the needs of the genome would dictate which functions persist.

In conclusion, our findings suggest retrocopied gene sequences represent a prevalent, recurrent, and rapid mechanism in primates and other organisms to evolve new genome defense functions including restriction of viruses. Although the presence of endogenous retroelements is probably net deleterious to the host, retrogene birth represents a mechanism whereby host genomes could nevertheless take advantage of the activities of these genomic pathogens to protect themselves against endogenous and infectious pathogens.

## Materials and methods

### Identification of *A3* retrocopies

*A3G* retrocopies were identified using BLAT of UCSC genome databases for marmoset (*Callithrix jacchus* draft assembly, WUGSC 3.2, GCA_000004665.1) and squirrel monkey (*Saimiri boliviensis*, sai-Bol1, GCA_00023585.1) with marmoset *A3G* (NM_001267742) as a query sequence. Additional copies were identified using BLASTn of the NCBI genome assemblies of Ma's night monkey (*Aotus nancymaae*, Anan_2.0) and capuchin (*Cebus capucinus imitator*, Cebus_imitator-1.0). The spider monkey retrocopy was identified using BLAST to query the NCBI HTGS database for reads from New World monkeys. See *Supplementary file 1* for detailed coordinates of each sequence.

### Mapping inactivating mutations in retrocopies

*A3I* sequences were queried using the codon-based and indel-sensitive alignment program LAST (http://last.cbrc.jp). The translated *A3G* sequence of *Callithrix jacchus* (NC_013914.1) was used as the reference sequence and indexed using the setting of 'lastdb -p -cR01' of the LAST aligner, and then the *A3I* sequences were queried using the setting of 'lastal -F15' to output in 'maf' format. The longest indel-sensitive translation of each *A3I* was then manually extracted from the maf output and aligned with mafft (https://mafft.cbrc.jp/alignment/software/) using the setting of '–anysymbol' to allow stop codons and frame shifting changes to be shown.

### Analysis of syntenic *A3G* retrocopies

Synteny of *A3G* retrocopies in marmoset and squirrel monkey was analyzed using UCSC table browser to download gene names within 1Mbp of either side of the retrocopy. Synteny was confirmed if the same gene was adjacent next to the retrocopy in both species. For five pairs of sequences that the tree suggested should be orthologous, we found shared genes on both sides of the retrocopies. For one retrocopy (*C5*), we found a shared gene on only one side of the retrocopies (*Figure 2—figure supplement 1*).

### Construction of *A3* phylogeny

*A3G* and *A3G* retrocopies were aligned using MAFFT (*Katoh and Standley, 2013*) with auto algorithm parameters within Geneious version 11.1.4 (*Kearse et al., 2012*). All retrocopies (both ORF-containing and retropseudogenes) were aligned using the complete alignable region defined by BLASTn. Trees were constructed using PHYML (*Guindon et al., 2010*) with NNIs topology search, BioNJ initial tree, HKY85 nucleotide substitution model, and 100 bootstraps.

### ORF retention simulation

To simulated the decay of retro *A3G* ORFs, we used the 'mutator' and 'orf_scanner' scripts developed by *Young et al., 2018*. The ORF of *Callithrix jacchus A3G* (identified from GenBank accession NC_013914.1, 1,150 bp) was used as the starting ORF. Combinations of several substitution, insertion, deletion rate and sexual maturation time were used for the simulation. We used substitution, insertion and deletion rate of $1.16 \times 10^{-8}$, $2 \times 10^{-10}$ and $5.5 \times 10^{-10}$ per site per generation for human (*Campbell and Eichler, 2013*), substitution, insertion and deletion rate of $5.4 \times 10^{-9}$, $1.55 \times 10^{-10}$ and $1.55 \times 10^{-10}$ per site per generation for mouse (*Uchimura et al., 2015*), and substitution rate of $8.1 \times 10^{-9}$ for Night monkey (*Aotus nancymaae*) (*Thomas et al., 2018*). Sexual maturation time of human, mouse and New World monkeys were estimated to be 25, 0.3 and 1–9 years (http://genomics.senescence.info; https://animaldiversity.org). Each run simulates the mutation of the starting ORF for 50 million years, and the simulations with each set of parameters were repeated 10,000

times. The number of ORFs that were still open and at the same length of the starting ORF were counted at every 50,000 years of each simulation.

## Analysis of selective constraints in *A3G* retrocopies

RELAX (*Wertheim et al., 2015*) was carried out using the Datamonkey webserver (*Weaver et al., 2018*) and a PhyML (*Guindon et al., 2010*) tree of the MAFFT (*Katoh and Standley, 2013*) aligned nucleotide sequences of the subset of retrocopies that encode an ORF longer than 250 amino acids in addition to the New World monkey *A3Gs* with or without human *A3G*. We defined the branches leading to the *A3Gs* as reference branches and all of the other branches as test branches. The above nucleotide alignment and PhyML tree were input into the CODEML NSsites model of PAML (*Yang, 2007*). To test for selection along branches, these same input files were input into the branch model of PAML. To test for significance of branches with apparent dN/dS < 1, we fixed that branch at dN/dS = 1 and calculated the likelihood of this tree.

## RNA-seq analysis for retrocopy and *A3G* expression

We searched the NCBI GEO and SRA databases (October 2018) with the keywords 'Callithrix', 'Aotus', 'Saimiri' and 'Cebus' to find existing RNA-seq datasets from these species. *Callithrix jacchus*, *Aotus nancymaae*, and *Cebus capucinus* are used, matching the available species where retrocopies of *A3G* were identified. For *Saimiri*, *Saimiri sciureus* RNA-seq was used, for which no genome sequence has been published, and the retrocopy analysis in the rest of the text analyzes *Saimiri boliviensis*. All RNA-seq datasets (*Supplementary file 2*) were queried using the default parameters of the 'blastn_vdb' tool of SRA toolkit (*Leinonen et al., 2011*) and the identified *A3Gs* and *A3G* retrocopies in this work as query sequences. RNA-seq reads hit by blastn_vdb were then processed with a custom perl (https://www.perl.org) script to only keep the reads that match the query sequence at 100% identity across the entire RNA-seq read and maps uniquely to only one of the queried retrocopies or *A3G*. Read that passed these filters were tallied and organized by species, tissue type, and the retrocopy or *A3G* copy they match.

## LINE-1 retrotransposition assays

LINE-1 retrotransposition assays were carried out as previously described (*Xie et al., 2011*). For the mouse ORFeus luciferase assays 25,000 HEK293T cells (ATCC Cat# CRL-3216, RRID:CVCL_0063) were seeded into each well of a 96-well clear bottom, white-wall plate. 24 hr later, each well was transfected with 200 ng pYX016 (CAG promoter driving mouse ORFeus LINE-1 with globin intron and luciferase reporter) or pYX015 (*Xie et al., 2011*) (JM111 inactive human LINE-1 construct which contains loss-of-function mutations in ORF1p of LINE-1) and pCMV-HA-A3 or pCMV-HA-empty (RRID:Addgene_32530). 24 hr post-transfection, transfected cells were selected with 2.5 µg/ml puromycin for 72 hr. Cells were lysed and luciferase substrates provided using the Dual-Glo Luciferase Assay System (Promega E2920). Renilla and firefly luciferase activity were measured using the LUMIstar Omega luminometer. Retrotransposition is reported as firefly/renilla activity to control for toxicity.

## Virus infectivity assays

Single-round HIV-1 infectivity assays were performed as described previously (*OhAinle et al., 2006*; *Yamashita and Emerman, 2004*). To produce VSV-G-pseudotyped HIV-1, 50,000 HEK293T cells (ATCC Cat# CRL-3216, RRID:CVCL_0063) were plated in a 24-well plate, and 24 hr later, co-transfected with 0.3 µg lentiviral vector encoding luciferase in the place of the nef gene (pLai3ΔenvLuc2 (*Yamashita and Emerman, 2004*), pLai3ΔenvLuc2ΔVif (*OhAinle et al., 2006*), 50 ng L-VSV-G, and 300 ng pCMV-HA-A3G or pCMV-HA-empty. All viruses were harvested 48 hr after transfection and filtered through a 0.2 µm filter. p24 *gag* in viral supernatants was quantified using an HIV-1 p24 Antigen Capture Assay (ABL Inc). Virus equivalents to two nanograms of p24 *gag* were used to infect 50,000 SupT1 cells (ATCC Cat# CRL-1942, RRID:CVCL_1714) per well in a 96-well plate in the presence of 20 µg/ml DEAE-dextran. Forty-eight hours after infection, cells from triplicate infections were lysed in 100 µl Bright-Glo luciferase assay reagent (Promega) and read on a LUMIstar Omega luminometer (BMG Labtech). A3A Western blots were carried out using Covance mouse HA.11 Clone 16B12 anti-HA monoclonal antibody (Covance Cat# MMS-101P-200, RRID:AB_10064068).

## Acknowledgements

We thank members of the Malik, Emerman, and McLaughlin labs for valuable discussions. We especially thank Janet Young for comments and suggestions critical for the generation of this manuscript, and Lily Wu for technical training and consultation on virus restriction assays. This work was supported by a Howard Hughes Medical Institute postdoctoral fellowship of the Helen Hay Whitney Foundation, a National Institute of General Medical Sciences (NIGMS) at the National Institutes of Health (NIH) K99/R00 Pathway to Independence Award (grant number GM112941) to RNM; National Institute of Allergy and Infectious Disease at the NIH R01 (grant number AI3092) to ME; grants from the Mathers Foundation and an NIGMS at the NIH R01 (grant number GM074108) to HSM. HSM is an Investigator of the Howard Hughes Medical Institute.

## Additional information

### Funding

| Funder | Grant reference number | Author |
| --- | --- | --- |
| National Institute of General Medical Sciences | GM112941 | Richard N McLaughlin Jr |
| Helen Hay Whitney Foundation | | Richard N McLaughlin Jr |
| National Institute of Allergy and Infectious Diseases | AI3092 | Michael Emerman |
| G. Harold and Leila Y. Mathers Foundation | | Harmit S Malik |
| National Institute of General Medical Sciences | GM074108 | Harmit S Malik |
| Howard Hughes Medical Institute | | Harmit S Malik |

The funders had no role in study design, data collection and interpretation, or the decision to submit the work for publication.

### Author contributions

Lei Yang, Conceptualization, Data curation, Software, Formal analysis, Validation, Investigation, Visualization, Methodology, Writing - original draft, Writing - review and editing; Michael Emerman, Harmit S Malik, Conceptualization, Supervision, Funding acquisition, Methodology, Writing - review and editing; Richard N McLaughlin Jnr, Conceptualization, Data curation, Formal analysis, Supervision, Funding acquisition, Validation, Investigation, Visualization, Methodology, Writing - original draft, Project administration, Writing - review and editing

### Author ORCIDs

Lei Yang https://orcid.org/0000-0001-9284-1744
Richard N McLaughlin Jnr https://orcid.org/0000-0003-0950-2253

### Decision letter and Author response

Decision letter https://doi.org/10.7554/eLife.58436.sa1
Author response https://doi.org/10.7554/eLife.58436.sa2

## Additional files

### Supplementary files

• Supplementary file 1. Sequence coordinates, orthology groups, and ORF retainment for *A3Gs* and *A3G* retrocopies.

• Supplementary file 2. Read counts of retrocopies across 98 New World monkey RNAseq datasets.

- Supplementary file 3. Codon-based and indel-sensitive alignment of primate *A3Is*. Stop codons and frame shifts were included in the alignment: star (*) represents a stop codon, slash (/) represents a frame shift caused by deletion, and backslash (\) represents a frame shift caused by insertion. Header of the sequences indicate the names of species and the NCBI accession numbers where the sequences are extracted from.

- Transparent reporting form

### Data availability

All data generated or analyzed during this study are included in the manuscript, supporting files, or publicly available databases as listed in the Supplementary files 1 and 2. Raw data files have been provided for Figure 3.

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
