## [Decision Letter]

[Editors' note: this paper was reviewed by Review Commons.]

**Acceptance summary:**

We appreciate the well-supported notion that retrocopying rather than de novo integration has contributed to the evolution of the APOBEC family, and judge that this manuscript represents a significant addition to the field.

---

## [Author Response]

We thank the reviewers and the editor for the insightful and thorough assessment of our manuscript. In this response to review letter, we have listed the original review and responded to each critique after it.

Reviewer #1 (Evidence, reproducibility and clarity):Yang et al. submitted a manuscript describing the detection of pseudogenes ("retrocopies") of APOBEC3 (A3) genes in primates. The evolutionary history and relationship to specific A3s was analyzed and speculated that the maintained A3 retrocopies had a functional role at least early in the evolution and some may have still now. Functional data on some of the expressed retrocopies are presented on L1 and HIV.The authors claim that "retrocopying expands the functional repertoire of A3 antiviral proteins in primates". While almost of the genetic findings were published recently (Ito et al., 2020), the authors should more clearly describe how their data differ or confirm the data of Ito et al.

We thank the reviewer for their helpful comments which have guided revisions to our manuscript. We have taken steps to clarify the dramatic differences between our work and the recent publication from Ito, Gifford, and Sato.

Foremost, we respectfully disagree with the reviewer that the genetic findings in our work were contained within the Ito, et al. manuscript. Using a computational screen of assembled mammalian genome, the Sato group catalogued the gain and loss of APOBEC3 genes during the evolution of mammals. They found a fascinating correlation between the dynamics of *A3s* and ERVs that formed the precis of the paper. From their genome-wide search for *A3s*, Ito et al. describe several retrocopies of *A3s* in two New World monkey species, one of which retains a full-length open reading frame, leading to the statement that this gene may be functional.

We note that the retrocopies found in the Ito et al. paper span only two of the more than 20 species in which we identify *A3* retrocopies. Further, as a result of the breadth of our search for *A3s*, we find additional retrocopies in the same two New World monkey species that were examined in the Ito et al. paper. Finally, our study also examined functional capabilities of these additional *A3s*. These differences are highlighted by reviewer #3 who writes that relative to Ito et al., our manuscript studies the phenomenon of *A3* retrocopies “more deeply both by in silico analyses and cell culture experiments.” Reviewer #3 also summarizes the most important difference in our studies – our work presents a “conceptual advance that the antiviral gene expansion has achieved not only via tandem gene duplication but also via gene retrocopying”.

Lastly, we want to point out that the findings of our manuscript and Ito et al., 2020 were made concurrently. Indeed, throughout the preparation process of this manuscript, we were both aware of each other’s findings and shared preprints with each other. Most of the participating journals in Review Commons have “scoop protection” mechanisms that typically extend 6 months after the publication of the first article (Ito et al. was published Jan 2020), and our article was first submitted to Review Commons on February 14, 2020. Therefore, we feel confident that the “no scoop” policy applies to the minimal overlap between our paper and that of Ito et al.

Nevertheless, we have modified the text to more clearly acknowledge the parallel finding of some New World monkey retrogenes in the Ito, et al. paper.

The functional data (Figure 6) are interesting, but in the current form not complete. The authors have to show protein expression in the transfected cells (A3, L1, HIV) and level of encapsidation into viral particles. In addition, please analyze if the retrocopies express cytidine deaminase active enzymes.

We thank the reviewer for this comment, and we have added a Western blot of the six long-ORF-containing retrocopies as Figure 6—figure supplement 1. In this blot (from early in the project), we detected protein production in 293T cells for 3/6 retrocopies. In later optimizations of subsets of this blot, we were able to detect expression of the marmoset *A3G* and the other two marmoset retrocopies (marmoset-2 and marmoset-4). Despite optimization attempts, we were unable to detect protein for one of the retrocopies that restricts HIV-1ΔVif (capuchin-*C1*). Unfortunately, at this time the included blot is the only one we have in which all 6 constructs are included on a single blot. Optimally, all 6 constructs would be side-by-side in a single blot with optimized conditions, and we are happy to complete this experiment as soon as we are able to return to our lab after the SARS-2 quarantine is lifted. However, we think the added blot shows that some of the retrocopies produce protein and the absence of detectable protein from capuchin-*C1* could suggest that this retrocopy is especially potent in its restriction function or an idiosyncratic problem with detecting this protein using Western blot analyses.

We have not previously tested our lentiviral particles for levels of encapsidation of protein from each retrocopy. The value we see in this experiment is in explaining why some of the retrocopies that are expressed in producer cells may not restrict in target cells. While we note that precedent in the literature suggests that *A3* proteins which restrict HIV-1ΔVif are invariably encapsidated, we would be happy to carry out this experiment when our lab reopens.

In response to the reviewer’s request to test deaminase activity for each retrocopy, we note that Figure 4 shows the intactness of the deaminase motif in each retrocopy. However, we feel that a description of the mechanisms of restriction of these retrocopies is not a major point of this paper and is beyond the scope of the current investigation.

Reviewer #1 (Significance):Minor advance compared to Ito et al., 2020.

We respectfully and rigorously disagree with this assessment. Please refer back to the reviewer’s first comment. We defer, again, to reviewer 3’s assessment that our work presents a “conceptual advance that the antiviral gene expansion has achieved not only via tandem gene duplication but also via gene retrocopying”. Moreover, we must point out that the Ito et al., 2020 paper was entirely computational; indeed, several retrogenes that could computationally be predicted to be “dead” were confirmed by us as having antiviral activity.

Reviewer #2 (Evidence, reproducibility and clarity):Summary:Yang et al. study the expansion of APOBEC3 (A3) cytidine deaminases genes in primates. Authors find A3 retrocopies in several lineages in primates using Blast searches. Some are old and some are species specific. Some have disablements and some have intact ORFs. Authors study their mode of evolution, expression and functionality. Authors have performed detailed analyses including functional analyses. Some A3 retrocopies are broadly expressed and some have retained ability to restrain retroelements. I agree with the authors that their data supports that retrocopying has contributed the turnover in the repertoire of host retroelement restriction factors. Authors show that some retrocopies have remained active for long periods of time and they still show that they can restrict retroelements/retrovirus. This work provides an interesting example of immune system diversification. This study of the A3 family of proteins that are part of the vertebrate innate immune system and the data supporting turnover of these kind of immune system genes is strong. The work underscores that this is a way immunity genes evolve and it has parallels in the evolution of the TRIM gene family of immune genes. I just have a few comments. I think the work can gain from analyzing some aspects of the data in more detail and presenting the big picture in a summary table, even if it is just supplementary.Major comments:1) A3I is in many species. Does this mean it was preserved (i.e., functional for a while)? For how long have disabling mutations been accumulating? Can we get a sense of that? Even for other retrocopies, do we have a sense of how recent has the pseudogenization been? If it is very recent that means that the gene was active until not long ago.

Our analyses suggest that *A3I* was born in the common ancestor of simian primates and pseudogenized before the *Catarrhini/Platyrrhini* split. It is possible that *A3I* was functional within this extended period (~12-15 million years), but the presence of a shared truncating stop codon amongst all simian *A3Is* suggests the gene was no longer full-length at the time of diversification of the simians. Instead, the simian LCA likely encoded an *A3I* with a predicted ORF of 261 codons; if this truncated ORF were functional, it was then further truncated/pseudogenized with additional frame-breaking mutations which follow the phylogeny of primates.

We estimated the timeline of pseudogenization of each retrocopy using the species distribution of each syntenic retrocopy. We also note that we find full-length ORFs in three retrocopies which have been retained for a period of time at least as long as the age of the last common ancestor of the four New World monkeys. These old but intact retrocopies motivated our simulations of ORF retention rates (Figure 5).

2) In the PAML analyses test could be performed to test if the rate of evolution that are higher or lower than 1 for particular genes are actually significantly higher or lower than 1 for the particular gene comparing the likelihoods of the modes with the given rate with the one with the rate fixed to 1. Is there enough power to do this?

We thank the reviewer for pointing out this omission in our analysis. We did perform these tests and find a significant p-value for two of the nodes p=0.058 and p=0.025 respectively). We have updated the legend for Figure 5—figure supplement 1 to incorporate these p-values.

3) It seems to me that the synteny data Figure 2—figure supplement 1 reveals they are derived from independent retroposition events and not duplications of segments because those would include flanking genes. Is this correct? Authors could comment on that.

Yes, we think that each retrocopy we show in Figure 2—figure supplement 1 is likely created via an independent retrotransposition event. We have clarified in the text that Figure 2—figure supplement 1 shows the genes used to establish synteny to support orthology of the retrocopies shared amongst multiple species and that each of these ortholog groups presumably originated via distinct retrotransposition events.

4) In Figure 5—figure supplement 1, I am not sure why orthologous genes are not grouped together in the phylogeny and why p is smaller than 0.05. How should that figure, and the probability be interpreted?

We thank the reviewer for their comments on this figure. First, the reviewer identified an error in the tree in which the branch labels for “night monkey-C2” and “night monkey-SS1” were inadvertently switched. The corrected tree now follows the pattern expected by the reviewer. Second, we employed RELAX to “determine whether selective strength was relaxed or intensified in one of these subsets relative to the other” (Wertheim, et al., 2014). In this case, the p-value corresponds to the finding that the retrocopies (test branches) show intensification of selection relative to the intron-containing *A3Gs* (reference branches).

We have modified Figure 5—figure supplement 1 and the associated text to more clearly explain the specific hypothesis test we report.

5) It would be good to have a summary table that summarizes what genes have support for past or current functionality (preservation for long time or recent pseudogenization, expression, purifying or positive selection, ability to restrict retroelements) and in what lineages.

We agree with this reviewer suggestion. We have added the additional information including the number of frame disrupting mutations as a measure of age, intactness, and ability to restrict retroelements to Supplementary file 1. Thanks to this suggestion, Supplementary file 1 now serves as the master table to summarize the analyses of each retrocopy.

Reviewer #2 (Significance):This work provides an interesting example of immune system diversification. Authors study the APOBEC3 family of proteins that is part of the vertebrate innate immune system and the data supporting turnover of these kind of immune system genes. The work underscores that this is a way immunity genes evolve and it has parallels in the evolution of the TRIM gene family of immune genes revealing patterns in the mode of evolution of immunity genes. The audience of this work will be people interested in evolution of immunity, arms races and gene diversification and all evolutionary biologists interested in adaptation. I work in the field of comparative genomics and molecular evolution.Reviewer #3 (Evidence, reproducibility and clarity):Summary:This manuscript by Yang et al. is a well-written, intriguing paper highlighting the evolutionary significance of the gene creation via "retrocopying". The authors investigated the expansion of antiviral A3 genes via retrocopy in Primates and found that A3G-like retrocopies have been generated repeatedly during primate evolution. A part of A3 retrocopies found in New World monkeys retained full length open reading flames and anti-lentiviral capacities. Interestingly, the spectrum of anti-retroelement activity of A3 retrocopies was different from the original (i.e., intron-containing) A3G gene in these species, suggesting the occurrence of the functional differentiation followed by gene amplification. However, one of the main findings that many A3 retrocopies are present in New World monkey is in-line to a previous report (i.e., Ito et al., 2020), and the experimental validations were based on the human (not New World monkey's) retroelements. Nevertheless, this study deeply investigated the possible importance of A3 retrocopies for the host defense system evolution both by in silico analyses and cell culture experiments. This study provides the findings that can potentially expand our knowledge on the evolutionary arms races between retroelements and the hosts.Major:To strengthen the impact of this work, it would be better to increase the numbers of retroviruses in which the anti-retroviral capacities are investigated. I understand that it is difficult to examine retroviruses or L1s that are colonized naturally with New World monkeys, but I suppose it is not so difficult to investigate a variety of representative retroviruses such as murine leukemia virus (MLV) or the reconstructed human endogenous retrovirus K (HERV-Kcon). This additional experiment would be helpful to highlight that the spectrum of anti-retroviral activity of A3 retrocopies is divergent from the original A3G gene in these species and strengthen the concept to be proposed by this study.

The reviewer raises a fascinating question about whether retrocopies might have different restriction abilities relative to the other *A3s* in a given species. First, we feel that showing activity against one pathogen is sufficient for our claim that some of the *A3* retrocopies have antiviral potential. Second, we discuss in the paper the idea that HIV-1 is not the actual target of these (or any) innate immune genes in New world primates. We argue that any other targets we might test would also be surrogates for the “true” target of these genes.

Specific:1) Since the authors found the expansion of "functional" repertoire of A3 retrocopies specifically in New World monkey, it would be better to rephrase the title as"Retrocopying expands the functional repertoire of APOBEC3 antiviral proteins in New World monkeys".

We thank the reviewer for this comment but point out that a large portion of our manuscript presents our work on primates outside the New World monkeys. The reviewer is correct to note that our finding of restriction activity is limited to New World monkey retrocopies, but we feel that the current title will attract a broader audience and reflects the broader relevance of this work.

2) It might be better to add a figure summarizing which A3 retrocopies in which species retain nearly full length ORFs. For example, how about making a figure like Figure 2A for all the four representative New World monkey species?

We agree. We have added the length of the longest ORF for each retrocopy to Supplementary file 1.

3) Figure 3. It would be helpful to clarify that which cell of the heatmap corresponds to the intact A3 retrocopies.

We have added labels to indicate the intact *A3* retrocopies and adjusted the legend accordingly.

4) Introduction, paragraph four. It would be better to replace the word "protected" with "escaped" because this retrocopy subset should include the ones that are intact but not functional.

Changed as suggested.

5) Introduction, final paragraph. It would be better to rephrase "the common ancestor of mammals" as "the common ancestor of placental mammals" because A3 gene is absent in Marsupial.

Changed as suggested.

6) Introduction, final paragraph. Please rephrase "ongoing" as "recently-occurred".

Changed as suggested.

7) Results, paragraph three. I checked the multiple sequence alignment in Supplementary file 3 and suspect that the codon (alignment) position of the shared premature stop codon is 261 (not 264).

We thank the reviewer for pointing out this discrepancy. We have revised the text to reflect the correct position of the shared stop.

8) Results paragraph three. I could not understand the meaning of the sentence "Intriguingly, one lineage-specific mutation…". Please specify the position of mutation which the authors mentioned (in Supplementary file 3 or Figure 1B).

This portion of the text refers to a reversion of a stop codon in the orangutan *A3I*; specifically, the stop codon shared in all simians acquired a second mutation that created a longer ORF in only this species. We have removed this sentence from the text for the sake of clarity.

9) Subsection “Retention of putatively functional NWM A3G retrogenes” paragraph fivePlease refer Figure 5—figure supplement 1 here.

Changed as suggested.

10) Subsection “Retention of putatively functional NWM A3G retrogenes” paragraph fivePlease say "Significant relaxed selection was not detected" rather than "Our analysis detected no relaxation…".

Changed as suggested.

11) Figure 5—figure supplement 1 indicates "p=0.015", but the authors regard it as "not significant"?

We thank the reviewer for pointing out this confusing wording. We employ RELAX to “determine whether selective strength was relaxed or intensified in one of these subsets relative to the other” (Wertheim, et al., 2014). In this case, the p-value corresponds to the finding that the retrocopies show intensification of selection.

We have modified Figure 5—figure supplement 1 to more clearly explain the specific hypothesis test for this p-value. We have also modified the text to clarify this point.

12) Subsection “Retention of putatively functional NWM A3G retrogenes” paragraph fivePlease here refer the data showing the claim "Instead, these A3G retrocopies have evolved more rapidly than…".

Changed as suggested; see previous point.

13) Did the authors perform the statistical test on the dN/dS ratio analysis? If so, please mention the result of the test.

Yes, we did. Please refer to reviewer #2’s major point 3.

14) It would be better to modify the phrase "show evidence of recurrent selection for functional innovation".

Changed as suggested.

Reviewer #3 (Significance):This study provides a conceptual advance that the antiviral gene expansion has achieved not only via tandem gene duplication but also via gene retrocopying.Compare to existing published knowledge.Although one of the main findings that many A3 retrocopies are present in New World monkey is in-line to a previous report (i.e., Ito et al., 2020), this study investigated the above finding more deeply both by in silico analyses and cell culture experiments.Audience.Evolutionary biologists and researchers in the field of viruses (particularly retroviruses including HIV-1) and transposable elements would be interested in this work.Your expertise.Bioinformatics, genome biology, viruses, and transposable elements.